# The Tacotron-Based Signal Synthesis Method for Active Sonar

**DOI:** 10.3390/s23010028

**Published:** 2022-12-20

**Authors:** Yunsu Kim, Juho Kim, Jungpyo Hong, Jongwon Seok

**Affiliations:** 1Department of Information and Communication Engineering, Changwon National University, Changwon 51140, Republic of Korea; 2Sonar System PMO, Agency for Defense Development, Changwon 51618, Republic of Korea

**Keywords:** active sonar, deep learning, signal synthesis, Tacotron

## Abstract

The importance of active sonar is increasing due to the quieting of submarines and the increase in maritime traffic. However, the multipath propagation of sound waves and the low signal-to-noise ratio due to multiple clutter make it difficult to detect, track, and identify underwater targets using active sonar. To solve this problem, machine learning and deep learning techniques that have recently been in the spotlight are being applied, but these techniques require a large amount of data. In order to supplement insufficient active sonar data, methods based on mathematical modeling are primarily utilized. However, mathematical modeling-based methods have limitations in accurately simulating complicated underwater phenomena. Therefore, an artificial intelligence-based sonar signal synthesis technique is proposed in this paper. The proposed method modified the major modules of the Tacotron model, which is widely used in the field of speech synthesis, in order to apply the Tacotron model to the field of sonar signal synthesis. To prove the validity of the proposed method, spectrograms of synthesized sonar signals are analyzed and the mean opinion score was measured. Through the evaluation, we confirmed that the proposed method can synthesize active sonar data similar to the trained one.

## 1. Introduction

Sonar stands for sound navigation and ranging, and refers to equipment or methodology that identifies the existence, location, and characteristics of an underwater target object. As water is used as a medium in which propagation proceeds, detection is performed using sound waves [1]. Passive sonar is a receiver-only system that detects vibrations originating from objects, such as the vessel’s engines and propellers themselves. Relatively, it is simple to design and inexpensive to build. However, it requires a vast amount of data to distinguish only the desired signal by receiving all signals from animals and other ships. On the other hand, active sonar detects echo signals which are radiated from the transmitter, reflected by targets, and returned to the receiver. Since the radiated signal has a preset frequency characteristic and matched filtering can be applied to improve the signal-to-noise ratio (SNR) with the knowledge of the transmitted signal, active sonar is promising for underwater target detection in spite of the reverberation [2,3].

Active sonar modeling refers to estimating a returned echo signal reflected by an underwater target. In general, an active sonar modeling system consists of a transmitter, a receiver, and a target, and the transmitter and the receiver are located in different places to perform radiation and reception [4]. Various studies have presented methods for the simulated generation of sonar data [5,6,7,8], one of which is a simulation module provided by the North Atlantic Treaty Organization (NATO) submarine research center. In the simulation module, the signal emitted by the transmitter is simulated with the target through statistical calculation [5]. It also produces a more realistic signal by providing a target fading effect between sensors as seen in real-world sea environment datasets. However, in simplifying the sonar equation, the modeled signal inevitably differs from the data collected in the real ocean. In [6], La Cour et al. developed a multi-everything sonar simulation (MESS), reflecting the reverberation and simplified sea environment. However, the MESS also failed to closely realize the real ocean data because simplified sea environment parameters were added to the existing sonar equation. In addition, a simulator of non-acoustic and acoustics (SIMONA) simulators generate signals that reflect contact states and reverberations, as well as target shapes, multipath fading, and waveform types [7]. For the full simulation, beamforming and reverberation calculations, which are required to be input to the matching filter module, play a major role in the realistic data generation. Therefore, studies on generating reverberation in real time only for bidirectional active sonar have also been conducted [8]. However, these mathematical modeling-based methods are limited in accurately simulating by vast and complicated underwater environments.

Meanwhile, with the rapid development of deep neural networks (DNN), a lot of interest and research has been conducted on the technology of generating complicated time series signals of variable length from simple text information in the speech synthesis area [9,10,11,12,13]. Representatively, WaveNet [9] is a voice signal synthesis model that presented a remarkable performance in audio sample generation. However, there is a limitation in that it is only used as a kind of vocoder that uses a mel spectrogram, which contains linguistic features of the desired voice, not natural language text, as an input. In addition, DeepVoice [10] is a method that replaced conventional text-to-speech (TTS) pipelines with DNN. However, the method is limited because the learning process is not an end-to-end system. Subsequently, a model of the encoder–decoder structure is proposed to improve synthesis performance, and the importance is calculated using a pre-trained hidden Markov model (HMM) to predict the vocoder parameters [11]. Furthermore, a Char2Wav [12] model is designed to enable end-to-end learning, but additional preprocessing is still required in that it predicts the Vocoder parameter. Finally, Tacotron [13] is an end-to-end TTS model that succeeds in training a linear spectrum of speech data in natural language text at once. It consists of an encoder, a decoder, and attention modules, showing high generation performance enough to be used in commercial applications, and is widely used as a basic structure of the TTS models.

Therefore, in this paper, we propose a signal synthesis method for active sonar using the Tacotron model. To achieve our goal, we modified several main blocks of the Tacotron model to be operated for sonar signal synthesis. Starting with the introduction of the related works in Section 2, we explain the proposed method in detail in Section 3. Through experiments, we verify the effectiveness of the proposed method in Section 4 and conclude in Section 5.

## 2. Related Works

### 2.1. Active Sonar Target Signal Generation Based on the Highlight Model

Active sonar modeling means simulating a reflected signal against an underwater target. When a pulse signal is emitted to an underwater object in a steady state, various types of reflective signals are generated due to factors such as hull, medium, structural characteristics, frequency of incident waves, and pulse width. The echo signal of an active sonar using high frequency is produced by the reflection of the object’s representation, along with several equivalent scattering inside, characterized by the spatial distribution of the object’s highlights. Simulating a sonar signal is to consider everything that may occur in this process of reflection. Entering each point that hits the target into the reflection tracking algorithm has an infinite number of cases, thus the concept of highlights that simulate the target as a series of points is introduced [14].

At long range, an underwater target is represented by a single point generated from a single highlight. However, at short range, the distribution of highlights needs to be properly expressed because the target can have a distribution characteristic that varies with time and angle. Assuming that target is a submarine, the concept of a spheroid-placed highlight is used. The concept of a spheroid-placed highlight discontinuously recognizes the surface of the target that varies with the angle of incidence of the highlights attached at a specific position. Figure 1 shows the concept of the corresponding highlight:(1)$$p_{b}\left(r,x\right)=\overset{N}{\underset{g=0}{\sum }}h_{g}\left(r_{g},x\right)*p_{i}\left(x\right)$$

Given the time delay of each highlight, the signal $p_{b}$ reflected on all the highlights of the target can be expressed as a sum, as shown in (1). The receiver is r, the target is x, and there are a total of *N* highlight points in a multi-highlight system, including short-range underwater targets. At this time, the object transfer function of each highlight is called $h_{g}$ and the incident signal is $p_{i}$. This highlight modeling is simple but widely used due to high environmental approximation.

### 2.2. The Tacotron for TTS Modeling

At long range, an underwater target is represented by a single point generated from a single highlight. However, at short range, the distribution of highlights needs to be properly expressed because the target can have a distribution characteristic that varies with time and angle. Assuming that target is a submarine, the concept of a spheroid-placed highlight is used. The concept of a spheroid-placed highlight discontinuously recognizes the surface of the target that varies with the angle of incidence of the highlights attached at a specific position. Figure 2 shows the concept of the corresponding highlight.

The encoder–decoder structure and attention mechanism are the core building blocks of the Tacotron. The <Text, Speech> pair consists of the input and output of the model, respectively. The input uses natural language raw text and, as an output, linear and mel spectrograms are generated, respectively. Finally, the spectrograms are reconstructed as a WAV audio file through post-processing. The encoder receives text data and outputs a kind of text embedding, a vector that best represents the meaning of the input text sequence. The embedded text vector is used as information for reference when the decoder sequentially generates audio samples.

In addition, attention techniques determine the importance of text embedding vectors used by decoders to generate audio sequences at each time step. In the recurrent neural network (RNN) based sequence-to-sequence (seq2seq) model, the vanishing gradient problem in which the information itself slowly disappears when it is located at the beginning of the sentence exists. However, the attention technique successfully alleviates the problem. With these advantages, the Tacotron became the cornerstone of the end-to-end TTS model.

## 3. Proposed Method

### 3.1. System Structure

Figure 3 presents the overall structure of the proposed signal synthesis method for active sonar. The entire system is largely divided into four stages: dataset configuration, preprocessing, signal synthesis, and post-processing. In this paper, the dataset configuration part used a highlight-based active sonar simulator for data generation because the amount of real ocean datasets is insufficient to train the proposed system model. However, this data generation part has to be replaced by real ocean data ultimately. The dataset generated in this way is converted into data to be an input of the model through the preprocessor, and the input is synthesized by the DNN model and outputs a corresponding linear spectrogram. By estimating the phases corresponding to the synthesized spectrograms using the Griffin–Rim algorithm, the synthesized waveform signal is reconstructed through post-processing.

Figure 4 compares the inputs and outputs of Tacotron models utilized in speech and sonar signal synthesis areas. The two Tacotron models are in common with yielding linear spectrograms corresponding to the provided inputs, whereas they are different in relevance with time-order dependency. In other words, the input of the TTS model for Korean synthesis combines 80 symbols in a time-ordered sequence, but the input of the proposed model for sonar signal synthesis consists of 14 parameters regardless of time order. Therefore, in order to achieve our goal of reflecting the difference to the Tacotron model, we modified several main blocks and the modifications will be explained in detail in the following section.

### 3.2. The Tacotron-Based Sonar Signal Synthesis Model

Figure 5 shows the structure of the proposed Tacotron-based sonar signal synthesis model. As the input of the model, the parameter values used in the configuration of the dataset are normalized to real numbers in the range of [0, 1] in the order of depth, transmitter, receiver, target coordinates, and pulse information. After that, it goes through an encoder network to extract and convert from parameter information necessary for signal simulation to information necessary for synthesizing a linear spectrogram. The information vector output of the encoder is input into the decoder and goes through a process of synthesizing an active sonar echo signal corresponding to the input marine environment parameters. The active sonar echo signal is sequentially synthesized through multiple steps using decoder RNN in the form of a spectrogram. In the decoding step, the attention RNN refers to the necessary information from the parameters input to the model when synthesizing the frequency coefficient of the corresponding time step.

#### 3.2.1. The Sonar Environment Parameter Embedding Layer

In the conventional TTS model, tokenization is performed in the process of converting natural language text into vectors. After converting the order in which the word appears within a preset word dictionary into a one-hot vector, the neural network can judge the meaning of the word within the sentence by itself through a text embedding layer. It is a more effective approach in that it estimates meaning specific to each task than conventional word embedding algorithms, such as count vectorization [15], bag-of-words [16], and term frequency–inverse document frequency (TF–IDF) [17,18]. Figure 6 shows the operation process of this text embedding layer. Although it appears to compute the context vector $h_{c}$ of the word as the matrix product of the weights $W_{in}^{c}$ and the one-hot vector c, the $h_{c}$ can be easily obtained by selecting the corresponding row of the weight $W_{in}^{c}$.

In this paper, however, the input of DNN used to synthesize sonar signals represents a series of numerical vectors of environmental parameter values such as depth, pulse information, transmitter, receiver, and target coordinates. Unlike text embedding layers that estimate only the meaning of words that exist within a dictionary, sonar environment parameter embedding layers are continuous numbers and the number of cases can be infinite. In addition, due to the nature of the one-hot vector, meaningless zero values occupying only space are filled as elements, but sonar environment parameter vectors are denser and have unique meanings for each element. Therefore, it is necessary to design a weight vector so that the meaning can be inferred individually according to each parameter.

The operation process of the sonar environment parameter embedding layer is depicted in Figure 7. Unlike the sparse text embedding vector c, the sea environment parameter vector requires processing as a dense structure. The ith element $s_{i}$ of the vector S is assigned a weight W and a bias $b_{i}$ to output $h_{i}^{s}$, which transforms the meaning of the element into the information needed for signal synthesis purposes with a offset dimension *n_embed*. By performing an operation, such as a fully connected layer for each element of the input, it becomes possible to convert a single number into a nonlinear context vector. This allows the proposed model to synthesize more realistic active sonar data.

#### 3.2.2. Attention Layer

The proposed model has an autoregressive structure that synthesizes variable-length signals in units of a specific number of frames and again uses them as input to the decoder cell to output frames of the next time step. The RNN-based seq2seq model [19] inputs an entire sequence, referencing one information vector output from the encoder equally across all steps, and iterating the process until an end-of-sequence (EOS) token appears. However, after the Transformer model [20] came out, an attention layer that acts as an intermediary between the encoder and the decoder was introduced. Although the seq2seq structure is used as it is in the decoder output of the model, the attention layer determines its important input features at that time step, helping to process and generate more flexible performance. In this paper, we also use a structure that introduces an attention mechanism to enable the extraction and processing of encoded information necessary to form a signal spectrum output at frame time *t*. A Bahdanau attention mechanism [21] was used as the method of calculating attention in the same manner as the Tacotron, and its configuration is as follows:(2)$$Q=S_{t-1},\;\;K=H,\;\;V=H$$
(3)$$e^{t}=W_{a}tanh\left(W_{q}Q+W_{k}K\right)$$
(4)$$a^{t}=softmax\left(e_{t}\right)$$
(5)$$c^{t}=Va^{t}$$
(6)$$S_{t}=D\left(concat\left(c^{t},I_{t}\right)\right)$$

We performed the importance calculation process by repeating up to the last frame generation point T with a total of three vectors: Q,K, and V, which mean queries, keys, and values. $S_{t-1}$ refers to the decoder cell’s hidden state at the point just before the point *t* and *H* refers to the encoder cell’s hidden states at all points in time. Similarly, three types of weights: $W_{a},W_{q},\;\mathrm{and}\;W_{k}$ correspond to attention values, queries, and keys and are calculated to obtain attention score values $e^{t}$. Furthermore, $e_{t}$ becomes the attention value $a^{t}$ via the softmax function, and computes a context vector $c^{t}$ that utilizes only important information from the encoded information vector via a dot product operation with V. Finally, the calculated $c^{t}$ is concatenated to the input $I_{t}$ of the current decoder D(x), resulting in $S_{t}$. In this way, determining how important information is in synthesizing signals plays a crucial role in improving synthesis performance.

#### 3.2.3. Positional Decoding

The biggest difference between speech synthesis and the proposed sonar signal synthesis is in time information. The text sequence, which is the input of the speech Tacotron model, is representative time series data, and the list of each word in the sentence affects each other a lot in order, which also directly affects the output speech spectrogram. However, the sonar Tacotron model simultaneously receives a number of sea environment parameters as input. The corresponding values have a profound effect on the output of each element, but the arrangement order of the parameters does not affect the output. This temporal mismatch causes confusion as the decoder of the model does not correspond to the input in yielding the output sequence sequentially. Therefore, in this paper, we add a term to the input of the cell under decoding to indicate at what point in the entire file the frame corresponds to so that the decoder can track the context of the output point. The added temporal term is expressed in the form of a normalized floating point of [0, 1], and each time point t is expressed in the order of frames rather than information in seconds; thus, t divides the total number of frames in the generation file by T and uses it as location information. This alleviates the problem of perception confusion between inputs and outputs of models that do not correspond to each other in time, as described above.

#### 3.2.4. Target Masked L1 Loss

The design of appropriate cost functions is essential for the optimization of DNN models. To design a cost function comparing the linear spectrogram of the model’s output speech signal with the actual one, the conventional method used mean absolute error (MAE), as shown in (7):(7)$$L_{total}=\frac{1}{T}\frac{1}{N}\overset{T}{\underset{t=0}{\sum }}\overset{N}{\underset{i=0}{\sum }}\left|o_{t}^{i}-\hat{o_{t}^{i}}\right|$$

Time information, i.e., the total number of frames, is T, the ith frequency spectrum coefficient of the tth frame output by the model is $\hat{o_{t}^{i}}$, and the frequency spectrum coefficient of the reference signal is set to $o_{t}^{i}$. The L1 distance of $o_{t}^{i}$ and $\hat{o_{t}^{i}}$ was averaged over the entire frame and coefficients as a loss value, which presented better performance than using mean squared error (MSE) [13].

However, as described above, the sonar signal synthesis model does not effectively pass time information in the decoding step. Using positional decoding to provide temporal information to decoders is only an auxiliary role and is not a fundamental solution. In addition, due to the nature of the sonar signal, the background noise or clutter occupies most of the time except for the target portion at a specific point in time, so it is necessary to design the cost function to focus more on reducing the difference from the original. To solve this problem, we propose a target-masked MAE. A frequency coefficient of the target signal is mainly larger than the magnitude of the background noise. We calculate a binary mask M that is 1 where the target is estimated, as shown in (8–11), and 0 where there is no target. We add $L_{maskedLinear}$ to the overall cost function $L_{total}$, which allows the energy to be compared only to the target locations through the element-specific product of the output value of the neural network model and the frequency spectrum of the original signal.
(8)$$\mu =\frac{1}{N}\overset{N}{\underset{i=0}{\sum }}s_{t}^{i}$$
(9)$$M=\{\begin{array}{c} 0,\;\;s_{t}^{i}\leq 2\mu \\ 1,\;\;s_{t}^{i}>2\mu \end{array}$$
(10)$$L_{maskedLinear}=\frac{1}{T}\frac{1}{N}\overset{T}{\underset{t=0}{\sum }}\overset{N}{\underset{i=0}{\sum }}\left|M⊙s_{t}^{i}-M⊙\hat{s_{t}^{i}}\right|$$
(11)$$L_{total}=L_{linear}+L_{maskedLinear}$$

## 4. Experiments

### 4.1. Dataset Configuration

As described in Section 1, the proposed method aims to synthesize more realistic echo signals but requires more than a certain amount of data due to the nature of the data-driven approach. Because it is difficult to collect large amounts of sonar data in practice, data generated by an active sonar simulator are used for training the proposed Tacotron model. Ultimately, this generated data should be replaced by real ocean data when the real data is sufficiently collected.

In order to generate highlight-based active sonar data introduced in Section 2, the active sonar simulator receives parameters, including the coordinates of the transmitter, target, and receiver, calculates the signal reflected on the target, and outputs it in the form of a waveform. The input parameters of the highlight-based active sonar signal generator are summarized in Table 1.

When the entire range and the step are set to 15,000 m and 10 m, respectively, the sound ray is tracked until the entire distance is reached by the interval by the set parameter. The tracking altitude is calculated by dividing the [−20,20] degree range set as the default value by 400, the number of indexes, as shown in Figure 8.

In the experiment of this paper, a dataset was constructed by changing a total of three parameters that have a noticeable influence on the characteristics of sonar signals: depth, pulse length, and pulse center frequency. The sound velocity profile used for data generation is presented in Table 2. Highlight points were set to 10 and Gaussian noise is added to generated sonar signals corresponding to 10 dB SNR. The total number of cases considered in this experiment for training the model is summarized in Table 3.

### 4.2. Experimental Settings

Instead of loading a single long signal file and entering the entire file, it divides into frames and goes through all processes such as processing, input, and training. This section describes all parameters used in the experimental process. It is divided into two categories, audio processing and DNN training, and consists of parameter names, numerical values, and parameter descriptions.
(a)Audio processing parameters.
num_mels: 80, the number of mel filters to obtain the mel spectrogram.num_freq: 1025, the number of frequency coefficients.frame_length_ms: 50 ms, the length of the frame.frame_shift_ms: 12.5 ms, the length of the shift between frames.
(b)Model training parameters.
Parameter embedding dimension (same as encoder input dimension): 256.Encoder output dimension: 128.Attention type: the Bahdanau attention.Attention dimension: 256.Decoder input dimension: 256.Decoder output dimension (meaning the final output dimension of the model): num_freq and num_mels.


### 4.3. Experimental Results

To evaluate the synthesis performance of the proposed model, we examined three aspects: comparing spectrograms, checking attention alignment, and measuring mean opinion scores (MOS). The evaluation was conducted using an untrained test file, and 10 were separated for each parameter.

(a)Spectrogram comparison.

Spectrograms of the original sonar data generated by an active sonar simulator and spectrograms of the synthesized signal according to changes in depth, pulse length, and pulse center frequency are presented in Figure 9, Figure 10 and Figure 11. As can be seen in the figures, a target echo signal is successfully synthesized in each parameter condition. The time of the signal, which means the distance of the target and its shape are, similarly synthesized to the original signal. However, attenuation of the background noise level, which is synthesized by simple repetition, is observed.

(b)Attention alignment.

In order to check the attention mechanism, we visualized the importance of parameters for synthesizing sonar signals. As shown in Figure 12, the high parameter importance resulted in the corresponding training cases. Thus, we confirmed that the attention mechanism for model training was normally operated.

(c)The MOS score.

In order to measure the subjective quality between the generated and synthesized data, we conducted an MOS test [22]. A total of five persons participated in this experiment, and each type of data, i.e., generated and synthesized data, are evaluated. The average score of the participants for each sea environment parameter is shown in Table 4.

As shown in Table 4, the MOS score of the signal synthesized by the proposed model is similar to the generated original sonar signal. From these results, it can be seen the sonar signal synthesized by the proposed model generates a signal similar to the trained signals.

## 5. Conclusions

In this paper, we proposed a Tacotron model based on DNN for active sonar signal synthesis. The proposed Tacotron-based sonar signal synthesis method is suitably modified for active sonar. It consists of three submodels: an encoder that turns the input vector into an information vector needed to simulate the environment, a decoder that sequentially generates output based on the received information vector, and an attention module that extracts and processes only the information needed at each point in time when decoding. To evaluate the proposed method, we performed spectrogram comparison, attention results checking, and MOS tests. Through the evaluation, we confirmed that the proposed Tacotron model successfully synthesized almost similar data used for training. Furthermore, the proposed Tacotron model can be improved using variable signal generation models, such as Tacotron2 [23], combined with the WaveNet Vocoder and Flowtron [24] from NVIDIA, but it remains to be tested in a future work.

## Figures and Tables

**Figure 1 sensors-23-00028-f001:**
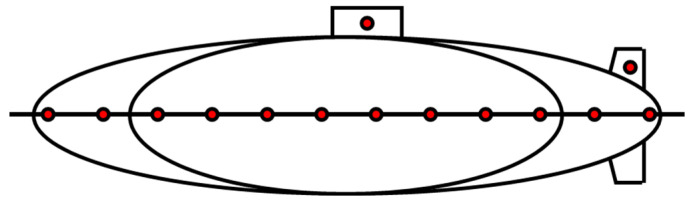
Spheroid-placed highlight modeling.

**Figure 2 sensors-23-00028-f002:**
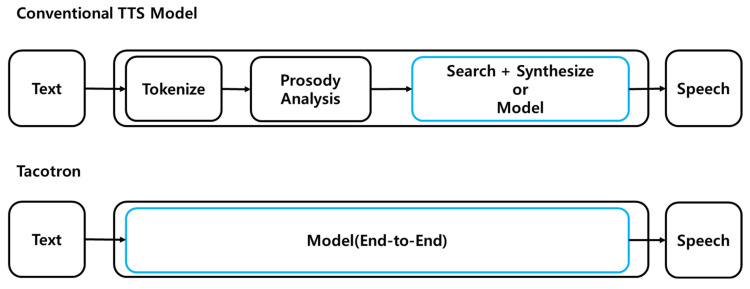
The Tacotron compared to conventional methods.

**Figure 3 sensors-23-00028-f003:**
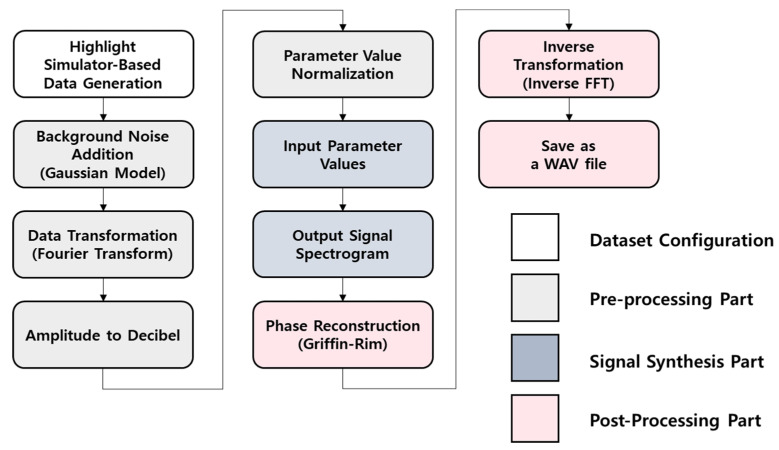
System structure.

**Figure 4 sensors-23-00028-f004:**
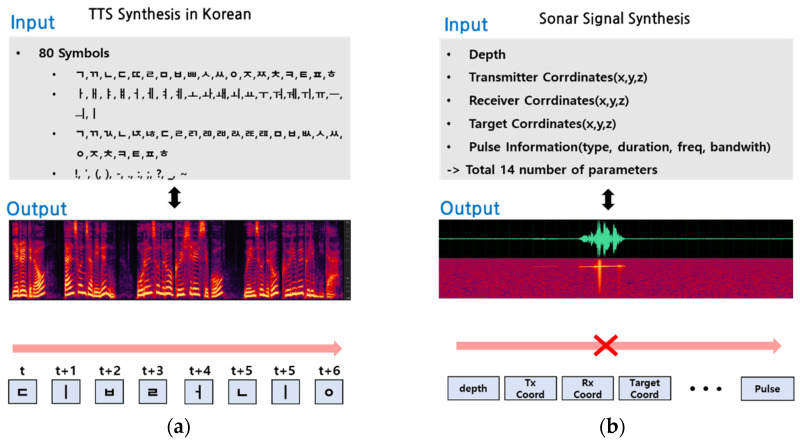
Input and output configuration. (**a**) The input/output structure of the TTS model. A total of 80 symbols are arranged in time order. (**b**) The input/output structure of the proposed sonar signal synthesis model. Fourteen marine environmental parameters are arranged regardless of time order.

**Figure 5 sensors-23-00028-f005:**
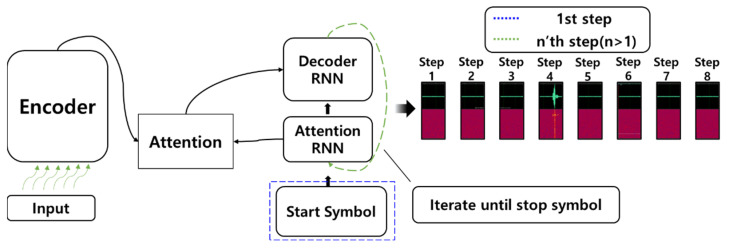
Model structure of the proposed sonar signal synthesis.

**Figure 6 sensors-23-00028-f006:**
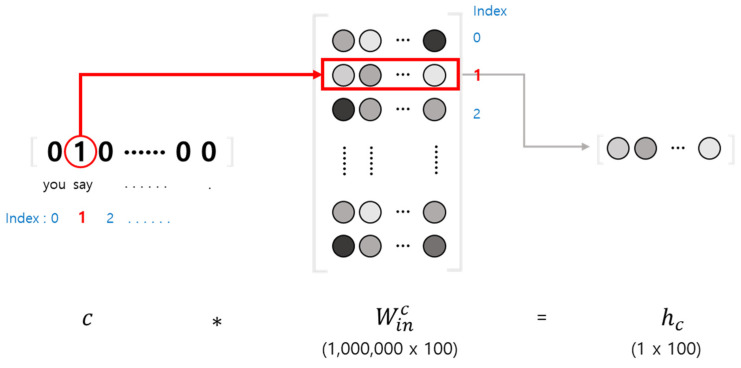
Word embedding layer.

**Figure 7 sensors-23-00028-f007:**
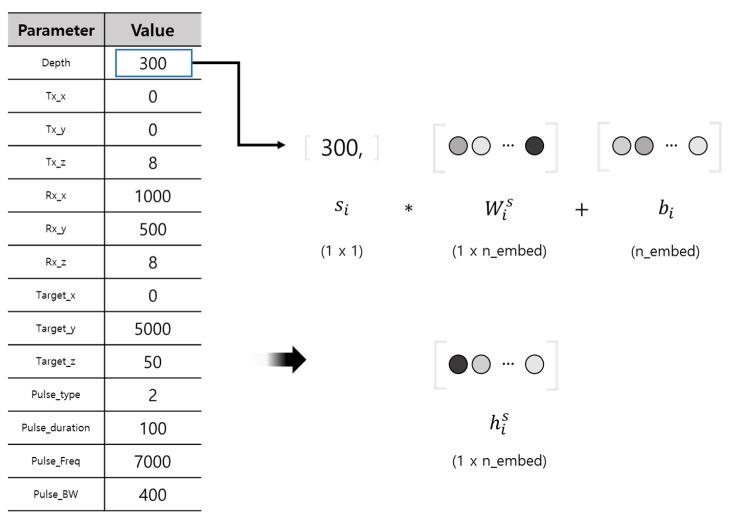
Sonar parameter embedding layer.

**Figure 8 sensors-23-00028-f008:**
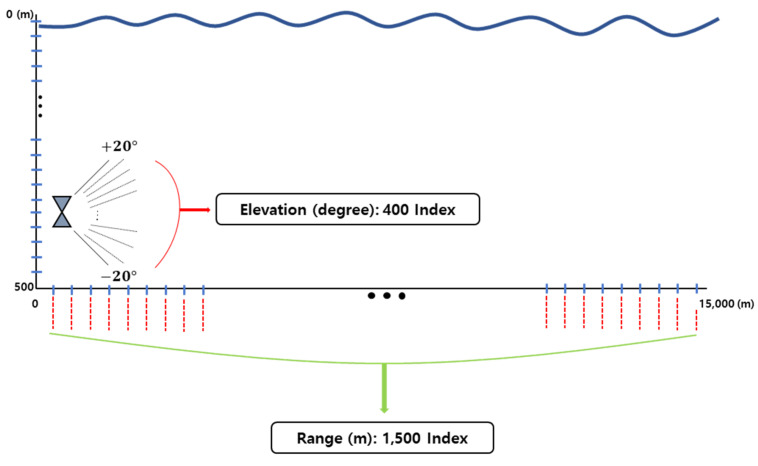
The ray tracing process.

**Figure 9 sensors-23-00028-f009:**
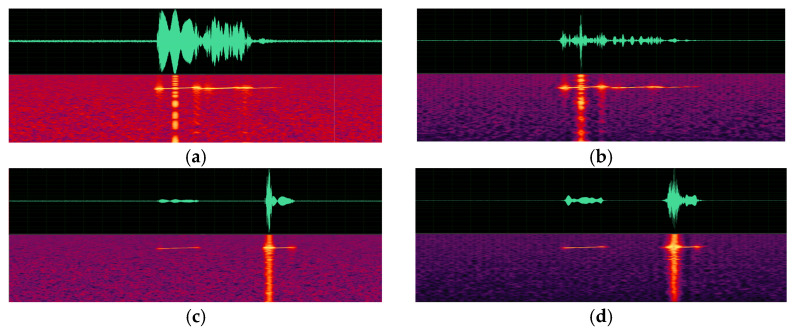
Spectrograms of active sonar signals with various depth parameters. (**a**) Original signal with a depth of 180 m. (**b**) A synthesized signal with a depth of 180 m. (**c**) An original signal with a depth of 660 m. (**d**) A synthesized signal with a depth of 660 m.

**Figure 10 sensors-23-00028-f010:**
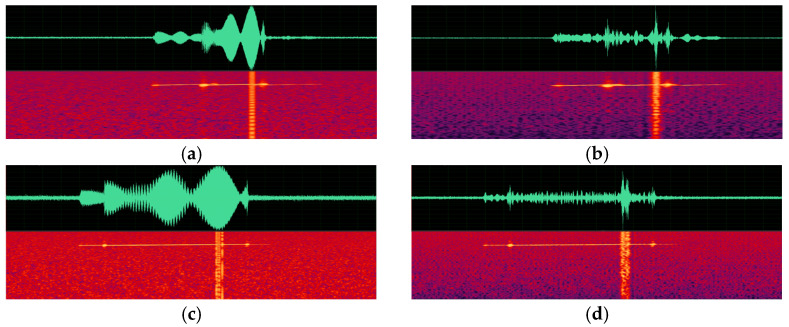
Spectrograms of active sonar signals with various pulse duration parameters. (**a**) An original signal with a pulse duration of 160 ms. (**b**) A synthesized signal with a pulse duration of 160 ms. (**c**) An original signal with a pulse duration of 770 ms. (**d**) A synthesized signal with a pulse duration of 770 ms.

**Figure 11 sensors-23-00028-f011:**
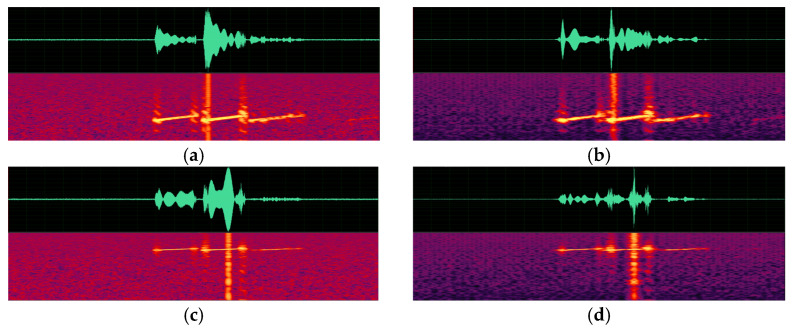
Spectrograms of active sonar signals with various pulse center frequency parameters. (**a**) An original signal with a pulse center frequency of 2080 Hz. (**b**) A synthesized signal with a pulse center frequency of 2080 Hz. (**c**) An original signal with a pulse center frequency of 5920 Hz. (**d**) A synthesized signal with a pulse center frequency of 5920 Hz.

**Figure 12 sensors-23-00028-f012:**
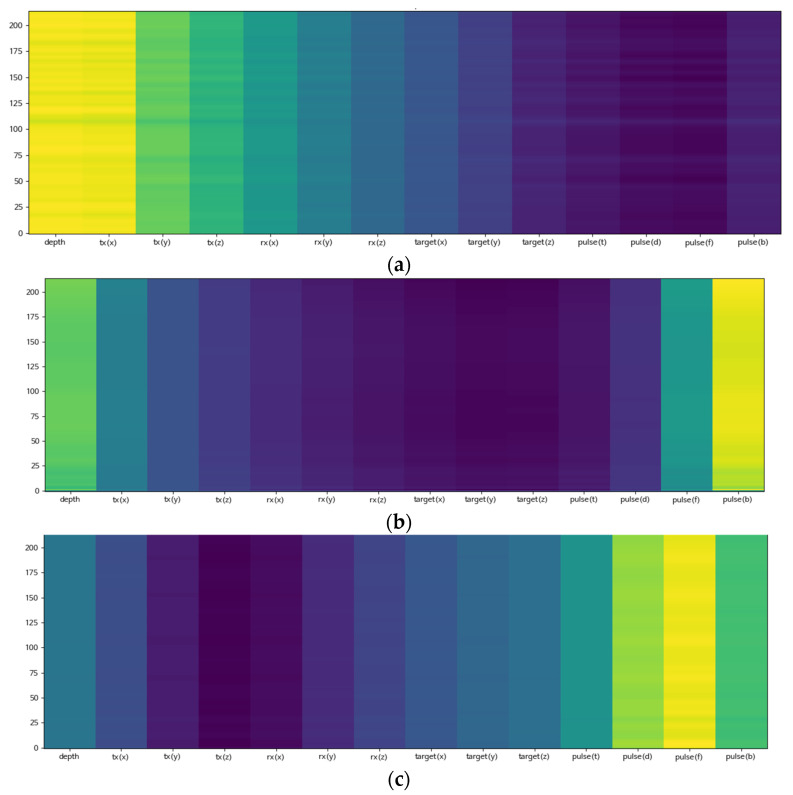
Attention alignments in the test. (**a**) Training only depth parameters. (**b**) Training only pulse center frequency parameters. (**c**) Training only pulse duration parameters.

**Table 1 sensors-23-00028-t001:** Input parameters of the highlight-based active sonar signal generator.

Parameter	Description	Setting Range
depth	depth information (unit: m)	[0, 5000]
tx	transmitter coordinates (x,y,z) (unit: m)	[0, range]/[0, range]/[0, range]
rx	receiver coordinates (x,y,z) (unit: m)	[0, range]/[0, range]/[0, depth]
target	target coordinates (x,y,z) (unit: m)	[0, range]/[0, range]/[0, depth]
svp	sound velocity profile (unit: m/s)	[0,∞]/[0,∞]
highlight	highlight information (number of points, relative position)	[0,∞]/[0,∞]
range	calculation limit range (min, max, and step) (unit: m)	[0,∞]
pulse	pulse information (type, duration, center frequency, and bandwidth) (unit: linear frequency modulation, and continuous wave/sec/Hz/Hz)	[0,∞]

**Table 2 sensors-23-00028-t002:** The sound velocity profile.

Depth (m)	Sound Speed (m/s)
0	1502
31	1504
60	1480
90	1477
120	1482
150	1481
180	1482
210	1476
240	1477
360	1479
⁝	⁝
5000	1480

**Table 3 sensors-23-00028-t003:** Dataset configuration.

Parameter	Step	Number of Files
depth	100–1000 m, 10 m per step	100
pulse duration	100–1000 m, 10 m per step	100
pulse center frequency	1000–7000 Hz, 60 Hz per step	100

**Table 4 sensors-23-00028-t004:** The MOS score.

	Depth	Pulse Duration	Pulse Center Frequency
Generated	4.03	4.10	4.51
Synthesized	3.68	3.78	4.32

## Data Availability

Not applicable.

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
