# Peer review of "The Tacotron-Based Signal Synthesis Method for Active Sonar"

_sensors, 2022, doi:10.3390/s23010028_

Round 1
Reviewer 1 Report
For solving the problem that the active sonar data for training the ML/DL models is insufficient, this paper proposed a new method for synthesizing signals, which modifies the Tacotron model to the field of sonar signal synthesis. The ideas presented were innovative, but there are still some problems in this paper:
1. Does the dataset configuration block in Figure 3 only reflect target highlight model, or also include the multipath fading and more environment impact? If this simulator was a method that was already proposed by others, the author may cite the relevant papers for explanation.
2. Line 171-173 said the input parameters included transmitter coordinates and else, but without environmental parameters. This conflicts with the description of Figure 4(b). And the back parts after encoder and the right part in Figure 5 is not explained, please add some explanation.
3. In Figure 7, how to calculate the weight W, and why does we need to add a bias here, which is different from the equation in Figure 6? Please explain it.
4. Line 259-260 didn’t explain the meaning of symbol l in .
5. Please explain why is calculated this way in equation (8).
6. In section 4.1, the author said “data generated by an active sonar simulator are used for training the proposed Tacotron model”. So did the author only use the simulated data for training or use both simulated data and real data?
7. In the introduction section, the author introduced some mathematical models. So what kind of method did the author use in the experiments for simulation to compared with the proposed method?
8. Line 307 said the noise was added at 10dB SNR. Does the performance of this model will change obviously at lower SNR?
9. It is suggested that some real data were used for testing the model in section 4.3(a), since the simulated data also have some difference with the real one. The synthesis accuracy of the proposed method cannot be accurately reflected only comparing with the simulated one. Authors can compare the spectrogram of real data, simulated data and the data generated by this proposed model using the same environment information, for analyzing their performance.
10. In 2.2, the introduction of the convential TTS model is only shown in Figure 2, but not in the paper. Figure 2 should be briefly explained, highlighting the comparison between the traditional TTS model and Tacotron used in the article, reflecting the advantages of Tacotron.
11. Please check whether the word ‘setp’ in line 297 is a wrong.
12. Table 4 does not give the unit of variable.
13. The experimental results in 4.3 in this paper are only compared with the echo signal of Tacotron and the transmission waveform. From the spectrogram, it can be found that the echo signal is indeed similar to the transmission signal, but it cannot be judged that the simulation results of Tacotron are correct. The results should be compared with the convential TTS model.
Reviewer 2 Report
Dear Authors,
Review of the article titled “Tacotron-based signal synthesis method for active sonar” (Manuscript ID: sensors-2089449) and submitted to Special Issue: "Recent Advances in Underwater Signal Processing" of “Sensors” to the section "Sensing and Imaging". The authors propose in this article a Tacotron model based on DNN for the active synthesis of the sonar signal. The proposed Tacotron based sonar signal synthesis method is suitably modified for active sonar. It consists of three sub-models: an encoder that transforms the input vector into an information vector needed to simulate the environment, a decoder that sequentially generates an output based on the received information vector, and a d attention that extracts and processes only the necessary information at each specific moment during decoding. To evaluate the proposed method, the authors performed spectrogram comparison, attention outcome verification, and MOS testing. Through this evaluation, they confirmed that the proposed Tacotron model successfully synthesizes almost similar data used for training and can be improved.
Overall the paper is well written and of interest. However, we note the absence of validated mathematical formulas describes the observed all physical phenomena and it should be justified. No experimental data or predictions from other calculations are available for comparison. The accuracy and validity the proposed model are therefore unclear. In consequence, the author needs to address the evidence before this reviewer agrees with publication of this paper. Also, a more careful literature review work is suggested. Hence the originality and novelty of manuscript (or the proposed the methods) are not clear.
This is a good paper, but you need to conduct a minor revision. After those corrections the manuscript may be published in the Journal. The following comments are split into some general ones and some more specific comment.
The comments are as in the attached PDF file:
